# Evidence to the Need for a Unifying Framework: Critical Consciousness and Moral Education in Adolescents Facilitate Altruistic Behaviour in the Community

**DOI:** 10.3390/bs12100376

**Published:** 2022-10-02

**Authors:** Gabriela Monica Assante, Nicoleta Laura Popa, Tudorița Grădinariu

**Affiliations:** Educational Sciences Department, Alexandru Ioan Cuza University, 700506 Iași, Romania

**Keywords:** critical consciousness, moral foundations, critical motivation, disadvantaged groups, adolescents, altruistic behaviour, community

## Abstract

Background: Critical consciousness represents an emancipatory pedagogical process whose central goal is developing the necessary skills to identify and act in the direction of changing social limitations. An important kind of action that helps challenge social limitations is altruistic behaviour. Moreover, moral values could enhance the effect of critical consciousness on altruistic behaviour. Method: This study aims to provide some empirical support for the benefits of a unifying framework between moral education and critical consciousness by exploring the association between critical motivation and moral foundations, and the moderating role of groups’ status (disadvantaged versus privileged) within this association. The present research explores the link between critical consciousness, altruistic behaviour, and the mediational role of moral foundations. The data was collected from participants studying in urban areas and disadvantaged rural areas. Hence, the socio-economic status of the individuals (disadvantaged groups versus privileged groups) is considered a moderator in this dynamic. The study sample comprised 1031 adolescents aged 13–19 (*M* = 16.51, *SD* = 1.54). Results: The findings emphasise that fairness and care moral foundations mediate the relationship between critical motivation and altruistic behaviour, and the moderator role of group status. In conclusion, the poor development of critical motivation in disadvantaged groups influences moral values development and, ultimately, affects individual behaviour in the community.

## 1. Introduction

Research in the field of critical consciousness (CC) primarily links to the study of disadvantaged groups, unequal systems and community conditions that perpetrate advantages for some categories of young people while creating disadvantages for others [1,2,3,4,5]. In this context, critical consciousness becomes an effective tool that provides individual aid in overcoming social disparities [6]. The critical consciousness concept is anchored in Freirean thought [7] and emerged at a turning point in Brazilian history (such as, the decolonisation process in Brazil). Since then, however, social contexts have changed; nowadays, there are still many marginalised groups whose development is affected by inequitable social conditions [1,2,3,8]. This theory relates to the Romanian context in the idea that the social school context enables social intervention by promoting management that excels in dialogue, participation, and emancipation, which leads to affiliative social behaviour and citizenship behaviour [9]. Critical consciousness is part of moral development and the result of various types of education, family education, life experiences or formal education [7]. The moral development of individuals evolves as they pass through different stages of life and formal education [10,11]. The way critical consciousness works to the benefit of the disadvantaged groups is by teaching them to critically analyse, reflect and act towards the unjust social conditions. This aspect is fundamental because marginalisation influences individuals to believe that their perspectives were insignificant, and that they are powerless in facing, changing, and overcoming social oppression. Therefore, as young people’s perception of social structures becomes more nuanced and complex, they become able to identify systemic inequities. Hence, their identities, moral beliefs, feelings of agency or alienation and civic behaviours tend to unfold differently. Feeling less constrained by their social conditions, developing a certain degree of agency and capacity to change these conditions results in overcoming developmental challenges and setting the course of their own lives [1,12].

On the other hand, it has been argued the crucial role of critical consciousness for those considered privileged, especially in identifying the reciprocal nature set between privilege and oppression [13,14]. Jemal [15] emphasises that for acquiring social change, privileged individuals need to learn to recognise social inequalities. Therefore, more privileged individuals would develop a critical consciousness about the oppression of others and will be able to identify the mechanism through which their privilege is preserved through the marginalisation of others [1].

Previous research has indicated a link between critical consciousness and civic engagement behaviours [3,11,16]. However, few studies focused on the relationship between critical consciousness and altruistic behaviour. Critical motivation is defined as the perceived moral commitment to addressing perceived inequalities. There is increasing interest in studying individuals’ incentives of altruistic behaviour because it offers valuable insight into the behaviour of future generations [17]. The conceptual formulations on altruism have changed from an altruistic personality type to personality traits that are adding to altruistic behaviour [18]. This approach was extended in three important directions. First, the understudied connection between critical motivation and altruistic behaviour was addressed. Second, the mechanisms linking critical motivation and altruistic behaviour were considered. Although the mechanisms linking critical consciousness and civic engagement were studied, no previous study has analysed moral values as mediators for the relation between critical motivation and altruistic behaviour. Third, both rural and urban school communities were considered in the same sample. The rural areas from which the data was collected were defined as such by the governmental authorities. Thus, were characterized by the relatively low density of inhabitants and buildings, the preponderance of natural landscapes, and the predominance of agro-pastoral economic activities. The rural areas from which data were collected have been for many years characterised by extreme poverty, being considered one of the poorest European regions [19,20]. In the studied socio-cultural space, the at-risk-of-poverty rate in rural areas is almost five times higher than in cities. Furthermore, the regional disparities are creating severe inequalities, raising barriers unfavourable to long-term sustainable growth [21]. Thus, was analysed whether the potential of critical motivation to predict altruistic behaviour is more fragile in disadvantaged groups. Several theoretical reflections and empirical studies on system justification [22,23] suggest that the people who feel powerless and dependent tend to internalize and behave based on motives that support the status quo, and are less likely to challenge inequalities, thus preserving long term psychological well-being. Moreover, the feelings of powerlessness and dependence among disadvantaged groups can feed system justification, although this relation may depend on contextual and individual factors. The fear for an uncontrollable world may lead to less opportunity for altruistic and prosocial behaviour, as people tend to ascribe negative traits to those who are less fortunate, and positive traits to the fortunate. Although a limited number of studies have been conducted on prosocial behaviours among Romanian children and adolescents, it is worthy to mention that previous research indicate that deprivation has a negative impact on prosocial behaviour [24], and female adolescents with better socio-economic background tend to display more generosity [25].

## 2. Literature Review

### 2.1. Critical Consciousness Structural Components

The process of developing critical consciousness is an emancipatory pedagogical one—the goal of Freirean education—and holds four major qualities. First, it involves raising awareness, which refers to the understanding that society can be developed and changed through human action. Second, it implies critical literacy, which covers the analytic abilities of thinking, reading, writing, speaking, and discussing, and can lead to discovering the profound meaning of any situation and applying that meaning to each context. The third quality is social de-construction by identifying and challenging various societal practices. Fourth, the self-education quality refers to the change of the school environment and society by introducing social change projects [26]. Therefore, critical consciousness is oriented towards helping individuals develop the necessary skills to identify and act against social inequalities. Its central educational objective is to trigger individual creativity and a constant, critical, reflexive ability that will further lead to critical transformative action in the social context [27].

Defined as the ability to engage in a reflective process regarding society’s prerogatives and action upon the world to transform it [7], critical consciousness includes three central components: critical reflection, critical motivation, and critical action. The core process in Critical Consciousness Theory is considered critical reflection because it implies a process of learning to identify social, political, and economic contradictions, and to act towards changing the restrictive conditions of social reality. Through the critical reflection process, individuals learn to put under scrutiny the elements and structures that lead to marginalisation. Critical reflection refers to the cognitive changes that happen as one recognises the roles of power and dominance in creating and maintaining systematic disparities between groups and the moral value, the unfairness, of such an arrangement. Therefore, critical reflection concerns the recognition that inequality and oppression are not moral, and that change is required [12].

The commitment and perceived ability to address such structures shape critical motivation or agency. The critical reflection process results in a social analysis and moral rejection of societal inequities such as the socio-economic, ethnic or gender inequities that limit individual agency and well-being. This process develops a systemic framework in which people examine social problems and inequalities. Critical action engages individuals to change perceived inequities. This entails individual or collective actions directed to change, challenge, and contest perceived societal disparities [28].

Critical motivation or critical agency refers to the individual’s perceived capacity to influence social change through their actions. It speaks of an individual’s perception regarding their ability and role in producing significant effects through individual actions to change social inequalities. This element has great value especially because it is more likely for people to engage in various actions if they have the feeling that this will lead to change [1]. Hence, critical motivation represents moral commitment to address perceived inequalities [28].

### 2.2. Critical Consciousness and Altruistic Prosocial Behaviours

Freire [7] stresses that when individuals engage in the process of social analysis, they feel compelled to act towards changing the unjust social conditions. Moreover, social analysis augments their understanding of the social issues [29]. Therefore, from the very first conceptual formulations, the development of critical consciousness has been linked to community action [7,30,31,32]. In this sense, a growing body of research is aimed at understanding critical consciousness and the roles it plays in youth development and community engagement [12].

Civic engagement behaviours vary from prosocial and altruistic behaviours to community engagement necessary behaviours. These behaviours represent the binder that currently weaves modern behaviours [33]. The need for individuals of all ages to become actively involved in any type of community or civic engagement behaviours is recognised as a source of personal meaning that often enhances individual well-being [34]. On this matter, there is a rising social need for teaching about the benefits and importance of helping others in a variety of diverse environments [35]. Other recent empirical evidence provides support for critical consciousness as a predictor for various positive outcomes related to young people’s development and well-being, such as vocational and career-related variables [2,36,37], academic achievement [14,38]. Therefore, because the components of critical consciousness can predict academic achievement and school performance, the development of CC in school would add value to both individual and community development [14,38]. Critical consciousness provides support for career development and vocational identity. By being more aware of societal barriers, students would be more equipped and motivated to overcome such obstacles [37]. In addition, they would achieve greater clarity regarding their vocational identity and more commitment to their future careers [36]. Moreover, critical consciousness provides support for increasing school connections and positive relationships with teachers [13].

There is increasing interest in studying individuals’ prosocial behaviour as well as the determinants of altruistic behaviour because it can provide valuable insight into the behaviour of future generations [17]. In psychological research, altruism is generally defined as individual behaviour that increases the well-being or the welfare of another individual, and therefore it is opposed by egoism [39]. In the same vein, Post [40] refers to altruists as persons doing something for the other’s sake rather than securing his or her well-being (p. 53), while Batson [41] (p. 606) emphasises the goal of altruistic behaviour, namely “increasing the welfare of one or more individuals other than oneself”. Recent studies add new evidence to the body of research underlining the positive connection of altruistic behaviour in adolescents with empathic concern and empathic skills [42,43,44], as well as emotional intelligence [45,46]. On the other hand, effects of altruism on adolescents’’ well-being are captured in studies highlighting that altruistic behaviour results in improved life satisfaction, as the ultimate outcome of higher school satisfaction and more positive emotional state [47,48]. 

From the beginning of the theoretical formulations, critical consciousness has been linked to cooperative social behaviour, i.e., for the benefit of the community. Freire [7] argues that underdevelopment of critical consciousness leads to hostile and individualistic behaviour among members of a community. Although previous studies have not explicitly investigated the link between critical consciousness and altruism, it is believed that altruistic behaviour is a good indicator of community behaviour that can occur at any needed time, especially in adolescents, and this research gap was intended to be addressed in the present study. The link between critical consciousness and civic engagement behaviours was previously explored in empirical studies [3,12,49] On the other hand, the literature suggests overlaps of altruistic prosocial behaviours and civic engagement: civic engagement may be expressed in authentic altruistic behaviours, but also in behaviours critical for a functional community life, such as political participation, which are not necessarily altruistic as they may also enclose a self-centered component [49,50].

Rushton [51] states that altruism is a universal value in every human society and concluded that an altruistic type of personality does exist. Later, however, research surprisingly changed this perspective: altruism is not regarded as a general factor of personality [51]; rather, personality traits are adding to altruistic behaviour. This relation differs according to the nature of the relationship between the people involved [52]. Therefore, altruistic behaviour towards family may be explained by kin selection [53], towards a friend or acquaintance implies the possibility of direct reciprocation [54], and towards a stranger was seen as a form of investment, the actors hoping on benefiting in the long-term by increased cooperation from others [55]. Furthermore, moral values and foundations are connected to commitment to various types of prosocial and altruistic behaviour. Because this type of behaviour increases during adolescence, its development is mainly associated with the development of moral foundations [56]. As previously mentioned, critical consciousness aims to raise awareness concerning issues of social equity; however, critical consciousness makes acknowledging equity-related situations easier, especially when those biases contrast with youth’s moral foundations [1,57]. Critical consciousness is linked to matters of moral reasoning and individuals’ moral values [58].

### 2.3. Critical Consciousness as a Moral Path

For challenging biases, critical consciousness is also linked to matters of moral reasoning being related to several aspects of the individuals, such as their moral values [42]. Therefore, aspects related to moral values and judgments are considered relevant in this dynamic. Several authors [57,58,59] have pointed out the importance of moral and identity development processes in facilitating critical consciousness, as young people consider their social positioning, moral values, and how they want to interact with the world when engaged in a specific action. Tyler [57] emphasises how critical consciousness facilitates the recognition of situations related to equity, especially when those inequalities conflict with the moral values related to the fairness of young people. Specifically given that, critical consciousness aims to raise awareness concerning those issues related to fairness and equity present in various social contexts [1].

Critical consciousness has a compounding effect on moral education. If the focus of moral education is to formulate general moral values and socio-cognitive skills, then critical consciousness is anchored in specific social and cultural aspects. Therefore, moral values and judgments are incorporated into critical consciousness development and learning processes. Critical consciousness can benefit moral education by shaping a reality-based perspective [59]. Even if its development relates to sensitive political contexts, the concept of critical consciousness is more current than ever. Nowadays, information literacy, reflection, and social action coherent with moral values are social glue.

Veugelers [59] appreciates that even if this connection is not often emphasised, the theory of critical consciousness can complement the theory of moral education by improving the perspective on justice (towards social justice). Thus, Freire [7] emphasises the importance of moral values such as fairness and social justice, and the power of the individual to control the circumstances of the social environment. Veugelers [59] acknowledges that both concepts are used by Freire [7] in a more political sense than in most research in the field of moral education and he cites Kohlberg whose approach to the moral value of justice/fairness is closer to an argument for equitable treatment, whereas social justice is more active and refers to the fight against inequity. Justice is rather focused on the action while social justice incorporates concern for the outcome and others. Social justice in the Freirean sense is embedded in social power relations. Veugelers [59] refers to contemporary authors who support Freire’s position on justice, preferring a politically embedded view of social justice at the expense of an abstract notion [60]. Mustakova-Possardt [61] considers critical consciousness as a moral pathway, both a moral and intellectual phenomenon, which is consistent with Freire’s [7] thought. Critical consciousness represents a way of interacting with the world impregnated with consequence. Therefore, developing the ability to reflect upon and question every aspect of the social, cultural, economic, and political contexts of reality includes, but cannot be reduced solely to a particular moral dimension. It also includes social actions and empowerment, and integrates intellectual, emotional, and moral aspects [61].

The literature on critical consciousness is continuously growing due to the recent availability of several scales designed to measure critical consciousness. These instruments intend to improve and consolidate the measurement of critical consciousness and its components to provide a deeper understanding of the conceptual ties [16,62,63]. Therefore, a more complex investigation of the relationship between critical consciousness components, altruistic behaviour and moral values remains justified using the appraisal methods explicitly designed to measure critical consciousness.

In this context, the first aim of this research was to explore the direct relationship between critical motivation and altruistic behaviour. To provide a better understanding of the aforementioned relationship and based on prior findings [57,59,61]. Veugelers, [59,61] that indicate moral values should be considered when exploring the link between critical consciousness and civic behaviour. Further, was explored the role that fairness and care for others’ moral foundations might play in the relationship between critical motivation and altruistic behaviour. These moral foundations were considered the closest ones to critical consciousness. Some theoretical assumptions also underlie the link between critical consciousness and fairness in terms of social justice, which incorporates both the fairness component and care for others [57,59,61].

Given the limited number of findings that explain the mechanisms linking critical motivation and altruistic behaviour, the second aim of the present study was to assess the mediating role of both fairness and care for others’ moral foundations as measured by Moral Foundations Questionnaire [64] and within the relationship between critical motivation and altruistic behaviour. Fairness and care for the others are the main foundations of individual morality, as defined in the moral foundation theory expanded by Haidt & Graham [65]. Fairness refers to the “valuation of individual rights and equality” and social justice (p. 104), while the care for the others is described as high sensitivity to sufferance, conjoined with kindness and compassion.

The third aim of the present study was to assess whether the mediator role of fairness and care for others’ moral foundations will be weaker for participants from the disadvantaged group. This research aims at forwarding the knowledge regarding the relationship between critical consciousness, specifically critical motivation, moral values, and altruistic behaviour in a sample of rural and urban adolescents. Hence, this study aims to provide some supporting evidence of how moral education and critical consciousness can complement each other and, subsequently, determine altruistic behaviour. By means of moderated mediation analysis, was aimed to investigate if some groups would indeed benefit more than others from such an approach.

The hypothesized moderated mediation model is depicted in Figure 1.

### 2.4. Hypotheses of the Present Study

Given the theoretical framework detailed in the above sections, the following hypotheses have been addressed:
**Hypothesis** **1.***Critical motivation is positively associated with altruistic behaviour.*
**Hypothesis** **2.***Critical motivation is linked to increased levels of fairness and care for others, which further amplify the altruistic behaviour.*
**Hypothesis** **3.***The relationship between critical motivation and moral foundations is significantly weaker for participants from the disadvantaged group.*

## 3. Materials and Methods

### 3.1. Participants

The participants were selected from the north-eastern region urban and rural schools of Romania. The information related to the study was disseminated to all the schools located in the mentioned area, but the data was gathered only from those which confirmed their willingness to participate to this research. The school administration disseminated the invitation to them to take part in the study. Participation was entirely voluntary. The sample consisted of 1031 adolescent students studying in secondary school or high school: 64% female and 36% male. The gender disparities are caused by the existing gender gap in the school enrolment rate. Hence, in the Romanian context, the female school population has an advantage over the male one [66]. Participants ranged in age from 13 to 19 years (*M* = 16.51, *SD* = 1.54). Participants studying in rural schools comprised 66.9% of the sample while remaining participants (33.1%) were studying in schools situated in urban areas. Participants from rural schools reside in disadvantaged rural communities. As previously noted, the region is one of the poorest in the European context, with considerably higher at-risk-of-poverty rate than urban areas, which determines both disparities and inequalities [51]. There were no exclusion criteria for the participants based on demographic variables. Participants’ characteristics are reported in the table below (Table 1).

### 3.2. Procedure

Parents of the participants were approached by the headmaster of each school. The headmaster was contacted by telephone and given information about the present study. Consequently, written information on the scope and procedures of the study was sent to all principals in the urban and rural schools from the north-eastern region. The school principals were the main link between the parents, participants, and the researcher. Therefore, the headmaster disseminated the information and asked the parents whether they were willing to consent participation of their children in the study or not. Moreover, written information regarding the scope and the objectives of the study was provided to the parents who agreed for their children to participate in the study; once signed, the headmasters collected the signed informed consent forms. The data was gathered by trained teachers that also provided information regarding privacy issues such as anonymity, voluntary participation including the possibility to withdraw from the study at any time. Data collection took place at the beginning of the school year, being administered in the form of a paper-and-pencil assessment. There were no missing data on the measured variables: critical motivation, altruistic behaviour, fairness, care moral foundations. The questionnaire took approximately 20 min to complete. The data were gathered during the first two months of the current year. This study was carried out following the recommendations of the Code of Ethics of the University. The protocol was approved by the Ethics Committee for Research of the Faculty of Psychology and Educational Sciences (No. 1997bis/03.03.2021). Following, the Declaration of Helsinki, all parents gave written informed consent for their children’s participation in the study.

### 3.3. Measures

The critical motivation measure was translated from English into Romanian using the forward–backward translation design [67]. Few corrections were made to the translations based on the back-translation process. The forward–backward translated version of the moral foundations measure is available on the Moral Foundations Questionnaire official Internet page (https://moralfoundations.org/questionnaires/, (accessed on 23 September 2022)). To assess altruistic behaviour, the Romanian-validated version of the self-report altruism scale distinguished by the recipient (SRAS-DR-RO) was used [28]. Prior to this study, the validity of the measures was explored by employing confirmatory factor analysis and the Cronbach’s Alpha reliability index. The sample size included 234 participants (*M*age = 19.25, *SD* = 1.09). The obtained coefficients for Critical Motivation (χ^2^ (2) = 4.34, *p* = 0.114, TLI = 0.93, CFI = 0.97, RMSEA = 0.07, Cronbach’s Alpha = 0.72), Care for other (χ^2^ (9) = 13.03, *p* = 0.161, TLI = 0.90, CFI = 0.94, RMSEA = 0.04, Cronbach’s Alpha = 0.75), Fairness (χ^2^ (9) 28.64, *p* < 0.01, TLI = 0.91, CFI = 0.94, RMSEA = 0.07, Cronbach’s Alpha = 0.71) and, Altruistic behaviour (χ^2^ (14) = 36.77, *p* < 0.01, TLI = 0.90, CFI = 0.94, RMSEA = 0.08, Cronbach’s Alpha = 0.87) indicated a good model fit.

Several controlling measures were taken to prevent the common method bias [referinta]. As several authors recommend [citare], item ambiguity was reduced by using the forward-backward translation method. Furthermore, participants’ anonymity was a priority, and they were assured that there are no right or wrong answers and that their answers should be as honest as possible.

#### 3.3.1. Critical Motivation

To measure critical motivation, the specific subscale of the Critical Consciousness Scale-Short Form (CCS-S) was used [63]. The critical motivation sub-scale includes four items that respondents, and example items include statements such as *“It is important to correct social and economic inequality*”, and “*It is my responsibility to get involved and make things better for society”*. Participants answered on a six-point Likert-type agreement scale ranging from 1 (strongly disagree) to 6 (strongly agree). This measure supports inquiry that elicits a more nuanced understanding of the pathways of critical consciousness development [63]. The Cronbach’s alpha coefficient for this sub-scale holds a satisfactory value of 0.74.

#### 3.3.2. Altruistic Behaviour

To measure altruistic behaviour, a validated version of the self-report altruism scale distinguished by the recipient (SRAS-DR-RO) was used [28]. This scale was formulated on evolutionary grounds and evaluates altruism in terms of the frequency of altruistic behaviours towards various receivers such as family members, friends, and strangers in everyday life [52]. It consists of 21 items rated on a five-point Likert scale from 1 (never) to 5 (very often). Example items for this dimension included *“I listened to the problems and dissatisfaction of a friend*” and *“I offered support to a family member when he was not feeling well”*, or *“I helped a stranger put his luggage on the train rack or in the bus hold*”. Cronbach’s alpha value of this measure is 0.89.

#### 3.3.3. Moral Foundations

The moral foundations of fairness and care for others were measured using the designated sub-scales from the short-form of the Moral Foundations Questionnaire (MFQ-S) [68]. This scale was translated into Romanian by a researcher and backtranslated into English by a professional translator (available at www.moralfoundations.org, (accessed on 23 September 2022)). Extensive cross-cultural research has been carried out using the MFQ questionnaire [68]. Both moral foundations sub-scales (fairness and care) include three items that assess the perceived relevance of moral concerns and three items that assess agreement with moral judgments. Participants responded to the relevance items on a Likert scale ranging from 0 (not at all relevant) to 5 (extremely relevant) and to the judgment items on a Likert scale ranging from 0 (strongly disagree) to 5 (strongly agree). The Cronbach’s alpha coefficients for the fairness and care scales are 0.73 and 0.71, respectively. Example items for each dimension included: “*Whether some people were treated differently than others*” or *“Justice is the most important requirement for a society”* (fairness moral foundation); *“Whether someone suffered emotionally*” or *“Compassion for those who are suffering is the most crucial virtue*” (care moral foundation).

### 3.4. Data Analysis

Preliminary data analyses were conducted to explore data normality. Data normality is examined in terms of skewness (SK≤ 3), kurtosis (KU≤ 10) [69]. Moderated mediation model was used due to its usefulness when the study aims to understand the reasons and under which conditions variables relate to one another. The combined model allows simultaneous investigation of contingent and indirect effects. Moderated mediation is necessary when mediation may not fit all groups of individuals [70]. To test the hypothesised moderated mediation model, the custom dialog PROCESS 3.5 (Model 8) for IBM SPSS version 24 for Windows was used [71]. This solution enables simultaneous testing of multiple mediators while providing bootstrap confidence intervals (CIs) for the indirect effects [71]. In addition, this protocol allows building bootstrap-based confidence intervals to test the significance of mediation effects in nonparametric and reduced biased conditions [72]. Hence, the bootstrap procedure holds the advantage that is that it does not assume normality, and adapts to more complex models [71]. In the present study, moderated mediation analysis was carried out using regression analysis and 5000 resamples (for estimating 95% confidence intervals). Gender and age were controlled in the model by including them as covariates. Further, even if it does not verify causation, correlation data was used because it can provide supplementary evidence when building an argument for causal claims such as a moderated mediational model. 

## 4. Results

### 4.1. Descriptive Statistics and Preliminary Analysis

Table 2 displays information related to Pearson’s correlations, Cronbach’s alpha reliability index, means, standard deviations, skewness, and kurtosis for the studied variables. The absolute values of skewness range from 0.005 to 1.067 (SK < 3) and the absolute values of kurtosis range from 1.187 to 0.159 (KU < 10), indicating that the data are normally distributed (Table 2). Critical motivation showed significant correlations with altruistic behaviour, fairness, and care in the investigated directions. Specifically, critical motivation is positively associated with altruistic behaviour (r = 0.24, *p* < 0.01), fairness (r = 0.39, *p* < 0.01) and care (r = 0.36, *p* < 0.01). Further, the link between critical motivation and altruistic behaviour, the mediational role of fairness and care, and the school location (rural/disadvantaged group vs. urban/privileged group) as moderator were explored.

### 4.2. Testing for Moderated Mediation

As anticipated, critical motivation positively predicted altruistic behaviour (total effect: b = 1.18, *p* < 0.01). Critical motivation also positively predicted the fairness moral foundation (b = 0.70, *p* < 0.01) and care moral foundation (b = 0.76, *p* < 0.01). Table 3 displays the complete results of the multiple regressions testing these effects.

The positive relation between critical motivation and altruistic behaviour was mediated by fairness and care moral foundations as indicated by significant indirect effects (for indirect effect through fairness, B = 0.13, SE = 0.06, 95% BCa CI: 0.092, 0.263 and for indirect effect through care, B = 0.22, SE = 0.06, 95% BCa CI: 0.111, 0.351) (see the upper side of Table 4). A post hoc power calculator was used to assess the statistical power of the indirect effect in the sample, the 20,000 Monte Carlo replications have shown 92% at *p* < 0.05 [73]. As was hypothesised, school location (rural vs. urban) conditioned the relation between critical motivation and altruistic behaviour (see lower side of Table 4 and Table 5). 

Overall, the results supported the hypothesised moderated mediation model (see Figure 2), revealing that the links between critical motivation and altruistic behaviour are mediated by fairness and care moral foundations. Further, the results only partially support the moderation hypothesis. The school location (rural vs. urban) that differentiates between the privileged and the marginalised groups in this study, significantly moderated the mediational effect of care moral foundation (see Table 5). Specifically, from lower (urban schools) to higher levels (rural schools) of the moderator, the interactional effect between critical motivation and school location was increasing. The effect on care moral foundation was strongest in urban schools, which led to greater values of altruistic behaviour. The analysis revealed that young individuals from disadvantaged areas score lower on critical consciousness and moral foundations.

## 5. Discussion

### 5.1. Theoretical Contributions

The present research examined the relationship between critical motivation, fairness and care moral foundations, and altruistic behaviour. Specifically, the present research explored whether the moral foundations (fairness and care) mediate the relation between critical motivation and altruistic behaviour. Ultimately, the moderating role of school location (rural vs. urban) in its relationship with critical motivation, moral foundations, and altruistic behaviour was examined. The rural areas from which the data were collected are among the poorest in European regions. Unfortunately, inequalities persist in the studied region, especially for people in rural and disadvantaged areas. Recent reports show how low living conditions affect school results, especially among children in rural and economically disadvantaged areas [21]. In this study, data from rural areas represent data from disadvantaged groups.

The results showed that critical motivation is positively associated with altruistic behaviour. Therefore, this study provides further empirical support to the assumption that a high level of critical consciousness is positively associated with civic behaviours. This results reflect previous findings that investigated in a different manner the relationship between these two concepts [57,59,61].

Furthermore, the results showed that the relationship between critical motivation and altruistic behaviour is mediated by both moral foundations—fairness and care. Therefore, these results agree with the previous theoretical formulations related to moral values and judgments. Previous theoretical formulations [57,59,61] have emphasised the importance of moral development processes in facilitating critical consciousness. Especially because when adolescents engage in a specific action with their environment, they consider their social positioning and moral values. Critical consciousness promotes the recognition of situations related to equity, especially when those inequalities conflict with the moral values related to the fairness of young people [57]; hence, critical consciousness aims to raise awareness concerning those issues related to fairness and equity present in various social contexts [1].

As hypothesised, school location moderated the relation between critical motivation, and both moral foundations and altruistic behaviour. Individuals from disadvantaged areas display lower moral commitment and moral development which are the foundations of civic behaviour, most probably under the influence of the self-perceived powerlessness and the tendency to pursuit system justification [23]. These outcomes may be deeply rooted in the educational disparities between urban and rural areas, still specific for the Romanian education system, largely determined by lower access to relevant educational resources and processes [74,75]. These results suggest that by unifying their efforts, moral education and critical consciousness could counter the negative effects of inequities and disparities that harm disadvantaged groups. By doing so, individuals from disadvantaged groups could use their agency to bring about positive change in the community. The paradox is that the people who would benefit the most from critical consciousness and moral development are the ones with the lowest levels. 

### 5.2. Practical Implications

The results indicate that equity, inclusion, and the quality of education remain important challenges. A recent report [21] shows that adolescents from disadvantaged areas showed no improvement in school performance. This may be determined by regional disparities such as in Romania, which are among the largest in the European Union. Poverty risks mainly affect rural areas and vulnerable groups and tend to be associated with a low level of education and an unfavourable socio-economic status. Furthermore, the results support the benefits of a unifying framework both for disadvantaged groups and for privileged groups in terms of civic behaviour.

The present study should inform the educational policies in the social education area as it provides empirical support for what other recent papers emphasised. That is, integrating critical consciousness development with moral education will enable students to apply moral understandings of harm, fairness, evaluate social norms and view themselves as moral agents in a community that works toward justice [76,77].

## 6. Conclusions

The present study provides evidence supporting the potential of critical motivation in determining civic actions. Moreover, both the fairness and care moral foundations proved important in explaining mechanisms for the relationship between critical motivation and altruistic behaviour. Due to the paucity of research on this topic, the results reported here bring important contributions that enhance understanding and support further research in this specific area of study. Interpreting the results of this study, several limitations should be noted: one type of civic behaviour from the broad spectrum of affiliative behaviours was included; the cross-sectional design did not enable any inferences regarding the causality of the relationships between critical motivation, moral foundations, and altruistic behaviour; variables that could act as moderators of moral foundations, such as personality traits or emotional regulation mechanisms, should also be considered to fully understand the mechanism through which critical consciousness and moral foundations interact in predicting civic behaviour; and the sample imbalance in terms of gender, with one-third male participants

Notwithstanding these limitations, several strengths should also be noted. First, in addition to previous research findings, the results support the relationship between critical motivation, moral foundations, and altruistic behaviour. Second, the mediating role of the moral foundations brings some clarity on the possible mechanisms through which critical motivation influences aspects of social behaviour such as altruistic behaviour. Third, the findings highlight the importance of a unifying framework of critical consciousness and moral education development in the case of disadvantaged groups.

## Figures and Tables

**Figure 1 behavsci-12-00376-f001:**
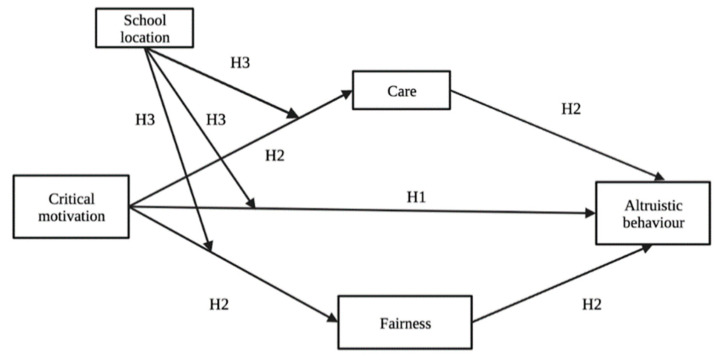
The Hypothesized Moderated Mediation Model.

**Figure 2 behavsci-12-00376-f002:**
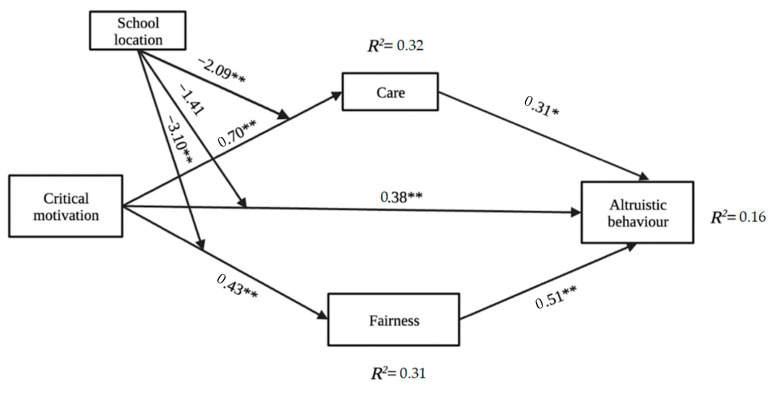
The moderated mediation model (*N =* 1031). Note: Unstandardized path coefficients are reported. * *p* < 0.01; ** *p* < 0.001.

**Table 1 behavsci-12-00376-t001:** Sample Demographic Characteristics.

Sample Characteristics	*n*	%	M	SD
Age			16.51	1.54
Gender				
Female	660	64%		
Male	371	36%		
School location				
Rural	341	33.1%		
Urban	690	66.9%		
School grade				
Eighth grade	94	9.11%		
Ninth grade	226	21.92%		
Tenth grade	196	19.01%		
Eleventh grade	213	20.65%		
Twelve grade	302	29.29%		

**Table 2 behavsci-12-00376-t002:** Descriptive Statistics, Reliability Estimates and Correlations Between Variables.

Variables		1	2	3	4
1.	Critical motivation	**0.74**			
2.	Fairness	0.399 ^**^	**0.73**		
3.	Care	0.362^**^	0.766 ^**^	**0.71**	
4.	Altruistic behaviour	0.241 ^**^	0.345 ^**^	0.367 ^**^	**0.89**
Mean		18.57	19.42	22.57	75.69
SD		4.10	5.08	5.97	14.73
SK		−1.067	−0.281	−0.363	−0.005
KU		1.187	−0.206	0.159	−0.470

Note: ** *p* ≤ 0.01 (two-tailed). Alpha Cronbach’s coefficients are shown on the diagonal (see boldface).

**Table 3 behavsci-12-00376-t003:** Results for the Regression Models used for Testing the Double Moderated Mediation.

	Coefficient	SE	*t*	*p*
*Fairness as outcome*				
Critical motivation	0.70	0.13	5.44	0.000
School location	−2.09	0.30	−6.86	0.000
Interaction 1	−0.17	0.07	−2.38	0.017
*Care as outcome*				
Critical motivation	0.76	0.15	5.00	0.000
School location	−3.10	0.35	−8.70	0.000
Interaction 1	−0.20	0.08	−2.35	0.018
*Altruistic behaviour as outcome (DV)*				
Critical motivation	1.18	0.42	2.81	0.005
Fairness	0.30	0.13	2.43	0.015
Care	0.48	0.11	4.72	0.000
School location	−1.41	1.01	1.39	0.163
Interaction 1	−0.45	0.23	−1.96	0.051

Note: Interaction 1 = product between critical motivation and school location.

**Table 4 behavsci-12-00376-t004:** Mediation and Moderation Effects Between Critical Motivation and Altruistic Behaviour.

	Coefficient	Standard Error	Confidence Interval	95%
			Lower Limit	Upper Limit
*Multiple mediation*				
Fairness	0.13 ^a^	0.06	**0.092**	**0.263**
Care *Moderated multiple mediation*	0.22 ^a^	0.06	**0.111**	**0.351**
Fairness	−0.05 ^b^	0.03	−0.140	0.005
Care	−0.09 ^b^	0.05	−**0.216**	−**0.010**

Boldface highlights significant mediation and moderated mediation effects. ^a^ Indirect effect of critical motivation on altruistic behaviour for each proposed mediator. ^b^ Index of moderated mediation for each mediator.

**Table 5 behavsci-12-00376-t005:** Conditional Indirect effects of Critical Motivation on Altruistic Behaviour at Values of the Moderator.

	Moderator Efficacy	b (SE)	95% CI
*Fairness moral foundation*			
Urban schools	0.16	0.08	(−0.005, 0.327)
Rural schools	0.10	0.05	(−0.003, 0.223)
*Care moral foundation*			
Urban schools	0.27	0.08	**(0.118, 0.447)**
Rural schools	0.17	0.05	**(0.075, 0.286)**

Boldface highlights significant moderated mediation effects.

## Data Availability

The data presented in this study is made available on request from the corresponding author.

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
