# Peer review of "Evidence to the Need for a Unifying Framework: Critical Consciousness and Moral Education in Adolescents Facilitate Altruistic Behaviour in the Community"

_behavsci, 2022, doi:10.3390/bs12100376_

Round 1

Reviewer 1 Report

The paper is about a very interesting topic. There are some problems for the publication (in my opinion) in the actual state of document:

  • Abstract needs to be clearer in order to briefly explain the research. I highly recomment to use the structure Background/Method/Results/Conclussions for abstract, especially because there is no explanation about the method (only about the sample). Also, another selection of keywords can be more productive for the paper (¿ altruistic behaviour?¿community?).
  • As general consideration, I highly recommen to write the papers in third person, I konow this point is right now in discussion but formally I suggest to write in this way.
  • Background is highly based in Theory of Paulo Freire, absolutey correct, but there is no connection with actual romanian social context in the evolution of the definition. In my opinion, it´s necessary to specify why this theory is the proper one for your social context (not for brazilian context) to introduce the study.
  • Methods part needs to explain the methodology of the study, why the authors have chosen that method, hhow the analysis were done, how you contrasted the information...
  • The authors decided to use corraltion data, it´s necessary to explain why the researchers decided it, if you checked the extracted information with other tools and what are the consequences of that choice for the validity of the data and the conclusions. more than this, lines 378-386 are clearly part of Methods.
  • In Results part there are dat from correlation, regression, mediation and moderation, with no explanation in Methods part about the use of these statistics. 
  • There are no explanations about how the authors defined the categories "rural and urban", adn they are fundamental for the research: size of the city? population? public consideration? legal conditions? Please, clarify
  • Results part is quite good, I really congratulate the authors for the presentation of data. 
  • Discussion, 430-433 the authors used the same arguments than in the introduction, they can be right but there are weak connections between this lines and the data included in the paper.

Author Response

September 29th, 2022

Behavioral Sciences 

            We want to thank you and the reviewers for the thoughtful comments and observations to our manuscript, entitled "Evidence to the need for a unifying framework: critical consciousness and moral education in adolescents facilitate altruistic behaviour in the community". We revised the manuscript in accordance to all the reviewers' comments. We will detail all the changes made, so that all our modifications can be easily identified throughout the text and the tables.

Reviewers 1 comments:

The paper is about a very interesting topic. There are some problems for the publication (in my opinion) in the actual state of document:

  1. Abstract needs to be clearer in order to briefly explain the research. I highly recomment to use the structure Background/Method/Results/Conclussions for abstract, especially because there is no explanation about the method (only about the sample). Also, another selection of keywords can be more productive for the paper (¿ altruistic behaviour?¿community?).
  • Thank you so much for you suggestion. The abstract and the keywords were modified as follows:

Background: Critical consciousness represents an emancipatory pedagogical process whose cen-tral goal is developing the necessary skills to identify and act in the direction of changing social limitations. An important kind of action that helps challenge social limitations is altruistic behaviour. Moreover, moral values could enhance the effect of critical consciousness on altruistic behaviour.

Method: This study aims to provide some empirical support for the benefits of a unifying frame-work between moral education and critical consciousness by exploring the association between critical motivation and moral foundations, and the moderating role of groups’ status (disadvantaged versus privileged) within this association. The present research explores the link between critical consciousness, altruistic behaviour, and the mediational role of moral foundations. The data was collected from participants studying in urban areas and disadvantaged rural are-as. Hence, the socio-economic status of the individuals (disadvantaged groups versus privileged groups) is considered a moderator in this dynamic. The study sample comprised 1031 adoles-cents aged 13–19 (M = 16.51, SD = 1.54).

Results: The findings emphasise that fairness and care moral foundations mediate the relation-ship between critical motivation and altruistic behaviour, and the moderator role of group status. In conclusion, the poor development of critical motivation in disadvantaged groups influences moral values development and, ultimately, affects individual behaviour in the community.

Keywords: critical consciousness; moral foundations; critical motivation; disadvantaged groups; adolescents; altruistic behaviour; community

  1. As general consideration, I highly recommend to write the papers in third person, I konow this point is right now in discussion but formally I suggest to write in this way.
  • Thank you so much for this observation. The entire manuscript was adjusted according to these indications.
  1. Background is highly based in Theory of Paulo Freire, absolutey correct, but there is no connection with actual romanian social context in the evolution of the definition. In my opinion, it´s necessary to specify why this theory is the proper one for your social context (not for brazilian context) to introduce the study.
  • Thank you for this observation. We added the following paragraph to bring some clarity over the role of Freire’s theory in our cultural context:

“This theory relates to the Romanian context in the idea that the social school context enables social intervention by promoting management that excels in dialogue, participation, and emancipation, which leads to affiliative social behaviour and citizenship behaviour (Barbu, 1998). Critical consciousness is part of moral development and the result of various types of education, family education, life experiences or formal education [7]. The moral development of individuals evolves as they pass through different stages of life and formal education [9, 10].”

  1. Methods part needs to explain the methodology of the study, why the authors have chosen that method, hhow the analysis were done, how you contrasted the information.
  • Thank you so much for this observation. The advantages of using moderated mediation models were included in the manuscript as follows:

“Moderated mediation models demonstrate usefulness when the study aims to understand the reasons and under which conditions variables relate to one another. The combined model allows simultaneous investigation of contingent and indirect effects (Edwards & Konold, 2020).”

  1. The authors decided to use corraltion data, it´s necessary to explain why the researchers decided it, if you checked the extracted information with other tools and what are the consequences of that choice for the validity of the data and the conclusions. more than this, lines 378-386 are clearly part of Methods.
  • Thank you for this observation. We have moved the lines in the Method section and have created a new Data analysis sub-section in which we explained the methodological choices. For explaining the use of correlational data we have added the following sentence:

“Even if it does not verify causation, correlation data was used because it can provide supplementary evidence when building an argument for causal claims such as a moderated mediational model.”

  1. In Results part there are dat from correlation, regression, mediation and moderation, with no explanation in Methods part about the use of these statistics. 
  • We have modified the Method and the Results section by adding the Data analysis section:

      3.4. Data analysis

Moderated mediation model was used due to its usefulness when the study aims to understand the reasons and under which conditions variables relate to one another. The combined model allows simultaneous investigation of contingent and indirect effects. Moderated mediation is necessary when mediation may not fit all groups of individuals.  (Edwards & Konold, 2020). To test the hypothesised moderated mediation model, the custom dialog PROCESS 3.5 (Model 8) for IBM SPSS version 24 for Windows was used (Hayes, 2013). This solution enables simultaneous testing of multiple mediators while providing bootstrap confidence intervals (CIs) for the indirect effects [60]. In addition, this protocol allows building bootstrap-based confidence intervals to test the significance of mediation effects in nonparametric and reduced biased conditions [61]. Hence, the bootstrap procedure holds the advantage that is that it does not assume normality, and adapts to more complex models (Hayes, 2009). In the present study, moderated mediation analysis was carried out using regression analysis and 5,000 resamples (for estimating 95% confidence intervals). Gender and age were controlled in the model by including them as covariates. Further, even if it does not verify causation, correlation data was used because it can provide supplementary evidence when building an argument for causal claims such as a moderated mediational model.

  1. There are no explanations about how the authors defined the categories "rural and urban", adn they are fundamental for the research: size of the city? population? public consideration? legal conditions? Please, clarify
  • Thank you so much for your observation. To bring some clarity we added the rural-urban definitions as follows:

The rural areas from which the data was collected were defined as such by the governmental authorities. Thus, were characterized by the relatively low density of inhabitants and buildings, the preponderance of natural landscapes, and the predominance of agro-pastoral economic activities.”

  1. Results part is quite good, I really congratulate the authors for the presentation of data. 
  • Thank you so much for your valuable feed-back.
  1. Discussion, 430-433 the authors used the same arguments than in the introduction, they can be right but there are weak connections between this lines and the data included in the paper.
  • Thank you for your observation. We included a referenced phrase in the Discussion section, justifying our view of the results, based on persistent educational disparities between rural and urban areas/schools in the Romanian educational system.

Reviewer 2 Report

I congratulate them for their work, which sincerely contributes to improving knowledge about civic and altruistic behaviour. Undoubtedly, the study has numerous strengths, but my role as a reviewer is to point out its limitations so that you can improve it. In this sense, I would like to point out the following aspects in case you consider that they can be improved. 

1. The references are relevant, but I have reasonable doubts about their timeliness and sufficiency. Only a quarter of them have been published in the last six years, and there is a lack of references to emotional intelligence in a study that deals with a topic such as altruistic behaviour that always requires empathy.

2. Their results show that school location moderated the relationship between critical motivation, moral foundations and altruistic behaviour, so that individuals from disadvantaged areas showed lower commitment and moral development. Now, if critical motivation represents the moral commitment to address inequalities, then why do they point out that these results may be due to the influence of self-perceived powerlessness and the tendency to seek justification in the system, rather than about the educational differences that might exist between one area and another, or to differential moral values according to the context or location of the school? 

3. The sample size is sufficient, but we do not know if it was established based on statistical power calculations, which should be pointed out or included among the limitations of the study or, better yet, its calculation should have been included a posteriori in the results.

2. The sampling procedure is not sufficiently described or is imprecise, so the sample is biased in terms of the sex of the participants.

3. The description of the procedure could be more precise. For example, it could be made explicit whether the people who collected the data had been trained to do so, or the type of information that the participants had about the objective of the study.

4. Finally, regarding the results, it would be necessary to know whether the data conform to normality tests and whether the statistical tests used meet their corresponding statistical assumptions.

Finally, I would like to congratulate you once again on your work and reiterate that my intention with these comments is to contribute to a better exposition and understanding of your work.

Author Response

September 29th, 2022

Behavioral Sciences 

            We want to thank you and the reviewers for the thoughtful comments and observations to our manuscript, entitled "Evidence to the need for a unifying framework: critical consciousness and moral education in adolescents facilitate altruistic behaviour in the community". We revised the manuscript in accordance to all the reviewers' comments. We will detail all the changes made, so that all our modifications can be easily identified throughout the text and the tables.

Reviewer 2

Comments and Suggestions for Authors

I congratulate them for their work, which sincerely contributes to improving knowledge about civic and altruistic behaviour. Undoubtedly, the study has numerous strengths, but my role as a reviewer is to point out its limitations so that you can improve it. In this sense, I would like to point out the following aspects in case you consider that they can be improved.

  1. The references are relevant, but I have reasonable doubts about their timeliness and sufficiency. Only a quarter of them have been published in the last six years, and there is a lack of references to emotional intelligence in a study that deals with a topic such as altruistic behaviour that always requires empathy (1.2.)
  • Thank you for your observation. Several very recent references have been added into the text, and the link between altruistic behaviour, empathic concern and skills as well as emotional intelligence in adolescence has been explicitly addressed.
  1. Their results show that school location moderated the relationship between critical motivation, moral foundations and altruistic behaviour, so that individuals from disadvantaged areas showed lower commitment and moral development. Now, if critical motivation represents the moral commitment to address inequalities, then why do they point out that these results may be due to the influence of self-perceived powerlessness and the tendency to seek justification in the system, rather than about the educational differences that might exist between one area and another, or to differential moral values according to the context or location of the school?
  • Thank you for your suggestion.Educational disparities between rural and urban schools in the Romanian educational system, as possible explanations for the results, have been described in the text in the Discussion section.
  1. The sample size is sufficient, but we do not know if it was established based on statistical power calculations, which should be pointed out or included among the limitations of the study or, better yet, its calculation should have been included a posteriori in the results.
  • Thank you for your recommendation. The Monte Carlo simulation was used to assess the statistical power as follows:
  • A post-hoc power calculator was used to assess the statistical power of the indirect effect in the sample, the 20,000 Monte Carlo replications have shown 92% at p < .05.”
  1. The sampling procedure is not sufficiently described or is imprecise, so the sample is biased in terms of the sex of the participants.
  • Thank you for your suggestion. The gender disparities were explained in the manuscript as follows:
  • “The gender disparities are caused by the existing gender gap in the school enrolment rate. Hence, in the Romanian context, the female school population has an advantage over the male one.”
  1. The description of the procedure could be more precise. For example, it could be made explicit whether the people who collected the data had been trained to do so, or the type of information that the participants had about the objective of the study.
  • Thank you for your recommendation. We added some clarity over the signaled issue as follows:
  • “Moreover, written information regarding the scope and the objectives of the study was provided to the parents who agreed for their children to participate in the study; once signed, the headmasters collected the signed informed consent forms. The data was gathered by trained teachers that also provided information regarding privacy issues such as anonymity, voluntary participation including the possibility to withdraw from the study at any time. Data collection took place at the beginning of the school year, being administered in the form of a paper-and-pencil assessment.”
  1. Finally, regarding the results, it would be necessary to know whether the data conform to normality tests and whether the statistical tests used meet their corresponding statistical assumptions.
  • Thank you so much for your recommendation. We have included in the manuscript the data normality indices as follows:
  • Preliminary analyses were conducted to assess data normality. Investigation of the normal distribution of data is examined in terms of skewness (SK ≤ 3) and kurtosis (Ku ≤10) (Kline, 2011). (…) The absolute values of skewness range from 0.005 to 1.067 (SK < 3) and the absolute values of kurtosis range from 1.187 to 0.159 (KU < 10), indicating that the data are normally distributed (Table 2).

Finally, I would like to congratulate you once again on your work and reiterate that my intention with these comments is to contribute to a better exposition and understanding of your work.

Reviewer 3 Report

Dear Authors,

Comment 1: The title should show the word "adolescents".

Comment 2: Abstract: The methods section should describe data collection and analysis methods.

Comment 3: I suggest that authors adjust the structure of Chapter 1, such as dividing the existing content into two chapters: Introduction, and Literature Review. The introduction part only introduces the research background, research significance, research gap, research objectives, research results, etc. For details, please refer to the prompts in the paper format template. The literature review, current research status, and hypothetical models should be moved to Chapter 2.

Comment 4: Hypotheses should be presented separately, as a single paragraph that would be clearer to the reader.

Comment 5: Figure 1 should be redrawn, keeping symmetry and aesthetics, and labeled H1, H2, etc.

Comment 6: Results: The issue of CMV should be addressed.

Comment 7: The square of R should be marked in Figure 2. Besides, Figure 2 also has the problem of not being aesthetically pleasing.

Comment 8: The Discussion structure is also suggested to be adjusted, and the content of this chapter should be divided into two branches: Discussion and Conclusion. The Discussion should be subdivided into two subsections: theoretical contributions, and practical implications.

Comment 9: Limitations should be more concise and moved to the Conclusion section.

Author Response

September 29th, 2022

Behavioral Sciences 

            We want to thank you and the reviewers for the thoughtful comments and observations to our manuscript, entitled "Evidence to the need for a unifying framework: critical consciousness and moral education in adolescents facilitate altruistic behaviour in the community". We revised the manuscript in accordance to all the reviewers' comments. We will detail all the changes made, so that all our modifications can be easily identified throughout the text and the tables.

Reviewer 3

Comments and Suggestions for Authors

Dear Authors,

Comment 1: The title should show the word "adolescents".

  • Thank you so much for your suggestion. The word 'adolescents' was included in the title

Comment 2: Abstract: The methods section should describe data collection and analysis methods.

  • Thank you so much for you suggestion. The abstract and the keywords were modified as follows:

Background: Critical consciousness represents an emancipatory pedagogical process whose cen-tral goal is developing the necessary skills to identify and act in the direction of changing social limitations. An important kind of action that helps challenge social limitations is altruistic behaviour. Moreover, moral values could enhance the effect of critical consciousness on altruistic behaviour.

Method: This study aims to provide some empirical support for the benefits of a unifying frame-work between moral education and critical consciousness by exploring the association between critical motivation and moral foundations, and the moderating role of groups’ status (disadvantaged versus privileged) within this association. The present research explores the link between critical consciousness, altruistic behaviour, and the mediational role of moral foundations. The data was collected from participants studying in urban areas and disadvantaged rural are-as. Hence, the socio-economic status of the individuals (disadvantaged groups versus privileged groups) is considered a moderator in this dynamic. The study sample comprised 1031 adoles-cents aged 13–19 (M = 16.51, SD = 1.54).

Results: The findings emphasise that fairness and care moral foundations mediate the relation-ship between critical motivation and altruistic behaviour, and the moderator role of group status. In conclusion, the poor development of critical motivation in disadvantaged groups influences moral values development and, ultimately, affects individual behaviour in the community.

Keywords: critical consciousness; moral foundations; critical motivation; disadvantaged groups; adolescents; altruistic behaviour; community

Comment 3: I suggest that authors adjust the structure of Chapter 1, such as dividing the existing content into two chapters: Introduction, and Literature Review. The introduction part only introduces the research background, research significance, research gap, research objectives, research results, etc. For details, please refer to the prompts in the paper format template. The literature review, current research status, and hypothetical models should be moved to Chapter 2.

  • Thank you so much for your recommendation. The structure of Chapter 1 was adjusted as suggested, and the whole text was re-organized accordingly.

Comment 4: Hypotheses should be presented separately, as a single paragraph that would be clearer to the reader.

  • Thank you so much for your suggestion. Hypotheses are presented now separately, in a dedicated subsection.

Comment 5: Figure 1 should be redrawn, keeping symmetry and aesthetics, and labeled H1, H2, etc.

  • Thank you so much for your observation. Figure 1 has been redrawn following the suggestions made by the reviewer.

Comment 6: Results: The issue of CMV should be addressed.

  • Thank you so much for your observation. We addressed the CMV problem as follows:
  • Several controlling measures were taken to prevent the common method bias [referinta]. As several authors recommend [citare], item ambiguity was reduced by using the forward-backward translation method. Also, participants' anonymity was a priority, and they were assured that there are no right or wrong answers and that their answers should be as honest as possible.

Comment 7: The square of R should be marked in Figure 2. Besides, Figure 2 also has the problem of not being aesthetically pleasing.

  • Thank you so much for your observation. Figure 2 has been redrawn following the suggestions made by the reviewer.

Comment 8: The Discussion structure is also suggested to be adjusted, and the content of this chapter should be divided into two branches: Discussion and Conclusion. The Discussion should be subdivided into two subsections: theoretical contributions, and practical implications.

  • Thank you so much for your suggestion. The structure of Discussion was adjusted as suggested.

Comment 9: Limitations should be more concise and moved to the Conclusion section.

  • Thank you so much for your observation. The text regarding limitation was modified/shortened and moved to the Conclusion.

Round 2

Reviewer 3 Report

Thank you for your revisions.